# U-Turn Diffusion

**DOI:** 10.3390/e27040343

**Published:** 2025-03-26

**Authors:** Hamidreza Behjoo, Michael Chertkov

**Affiliations:** Program in Applied Mathematics, Department of Mathematics, University of Arizona, Tucson, AZ 85721, USA; hbehjoo@arizona.edu

**Keywords:** generative models, diffusion, statistical physics

## Abstract

We investigate diffusion models generating synthetic samples from the probability distribution represented by the ground truth (GT) samples. We focus on how GT sample information is encoded in the score function (SF), computed (not simulated) from the Wiener–Ito linear forward process in the artificial time t∈[0→∞], and then used as a nonlinear drift in the simulated Wiener–Ito reverse process with t∈[∞→0]. We propose U-Turn diffusion, an augmentation of a pre-trained diffusion model, which shortens the forward and reverse processes to $t\in [0\to T_{u}]$ and $t\in [T_{u}\to 0]$. The U-Turn reverse process is initialized at $T_{u}$ with a sample from the probability distribution of the forward process (initialized at t=0 with a GT sample) ensuring a detailed balance relation between the shortened forward and reverse processes. Our experiments on the class-conditioned SF of the ImageNet dataset and the multi-class, single SF of the CIFAR-10 dataset reveal a critical Memorization Time $T_{m}$, beyond which generated samples diverge from the GT sample used to initialize the U-Turn scheme, and a Speciation Time $T_{s}$, where for $T_{u}>T_{s}>T_{m}$, samples begin representing different classes. We further examine the role of SF nonlinearity through a Gaussian Test, comparing empirical and Gaussian-approximated U-Turn auto-correlation functions and showing that the SF becomes effectively affine for $t>T_{s}$ and approximately affine for $t\in [T_{m},T_{s}]$.

## 1. Introduction

The fundamental mechanics of Artificial Intelligence (AI) involve a three-step process: acquiring ground truth (GT) data, modeling them, and then predicting or inferring based on the model. The core component of the prediction phase of generative AI is the generation of synthetic data.

The success of a generative AI model depends on how effectively it embeds information about synthetic data. In the emerging landscape of generative AI, this is achieved by leveraging the rich structure within these models. In particular, score-based diffusion (SBD) models [1,2,3] have emerged as a highly successful paradigm, currently representing the state of the art in image generation and competitive in other applications. SBD models excel due to their inherent structure, which consists of forward and reverse stochastic dynamic processes in an auxiliary time, enabling them to extract and redistribute information over time.

Remarkably, the principles underlying SBD models date back to earlier ideas from Stochastic Differential Equations (SDEs) by Anderson [4]. These principles allow for the design of a reverse process where the marginal probability distributions of images in the forward (noise-adding) and reverse (denoising) processes are identical by construction, and moreover preserve a detailed balance (DB) relation between the forward and reverse processes. This theoretical construct serves as a guiding principle for SBD models, enabling the redistribution of information from GT data, which forms the initial condition for the forward SDE. Over time, this information is encoded into the score function (SF) representing the re-scaled drift term in the reverse SDE, facilitating the generation of high-quality SD. Neural Networks (NNs) are employed to fit the exact SF, derived from Anderson’s work, for efficiency, avoiding memorization, and providing smoother representations dependent on the GT data.

### 1.1. Our Contributions

The concept of the U-Turn diffusion augmentation of a pre-trained SBD model arises from the intuition that the success of the SBD methodology is due to the reverse Stochastic Differential Equation (SDE) containing all necessary information to generate synthetic data. It also stems from addressing key questions essential to refining this approach: How is information about the GT samples distributed over time within the SF, which represents the drift term in the reverse SDE? Can we rigorously shorten the theoretically infinite time interval while preserving Anderson’s relations between statistics of the forward and reverse processes? How does the nonlinearity of the score function evolve over time? In the remainder of this subsection, we detail how this manuscript addresses each of these questions.

We investigate diffusion models of generative AI, focusing on generating new synthetic images from a probability distribution representing GT samples. A typical diffusion model consists of a forward noise-injecting phase, modeled as a linear Wiener–Ito stochastic process, which allows for analytical computation, and a reverse denoising phase, also a Wiener—Ito process but with a nonlinear drift term, requiring simulation to produce synthetic images. Our work centers on understanding how information about the GT samples, used to initiate the forward process, is encoded in the SF, which is approximated by an NN. The SF is crucial because it links the reverse process to the forward process: it is defined as the gradient of the log of the marginal probability distribution in the forward process at each point in time and is subsequently used as the drift term in the reverse process.

The main observation and contribution of this manuscript is that the essential information about the GT samples is primarily encoded in the SF during the initial stages of the forward process. This insight leads us to propose the **U-Turn diffusion** model (The original version of this manuscript was reported on arXiv in August 2023), which revises the original diffusion model by shortening both the forward process and the subsequent reverse dynamics. In this paper, we introduce a comprehensive theoretical analysis under the Gaussian assumption, which deepens our understanding of the U-Turn diffusion process. Additionally, we expand our experimental evaluation to include a broader range of datasets compared to our previous arXiv submission, thereby demonstrating the robustness and generality of our approach. In the U-Turn model, the reverse process becomes a detailed-balance conjugate of the forward process, requiring it to start from a sample of the marginal probability distribution at the forward process’s final, ‘U-Turn’ time, where the forward process is initialized with a GT sample. Notably, this construction allows for the creation of the U-Turn model directly from the original diffusion model’s score function without the need for retraining. While the U-Turn algorithm itself is described in Section 4, it is also prepared via a preliminary discussion, which can be viewed as a technical introduction to diffusion models, presented in Section 2.

Our second contribution, reported in Section 3, is the development of a toolbox of tests—the Kolmogorov–Smirnov Gaussianity test, SF amplitude test, and U-Turn auto-correlation tests—to determine the appropriate U-Turn time. These tests complement the standard Fréchet Inception Distance (FID) tests used to evaluate the quality of the diffusion models.

Third, and as reported in Section 4, we test the U-Turn scheme, as well as the aforementioned toolbox of tests on ImageNet datasets. Our experiments with the pre-computed/trained SF of the class-conditioned ImageNet dataset [5] led to the discovery of a critical U-Turn time, $T_{m}$, which we refer to as the **Memorization Time**. Specifically, if the U-Turn time, $T_{u}$, is shorter than $T_{m}$, the new sample generated is close to the original GT sample used to initialize the reverse process at $T_{u}$; however, if $T_{u}>T_{m}$, a truly new sample is produced. We observe that $T_{m}$ varies across different classes within the ImageNet dataset. The schematic illustration of what U-Turn discovers in the case of a single-class score function is illustrated in Figure 1a.

Fourth, we conduct, in Section 5, experiments with the multi-class CIFAR-10 dataset and its pre-trained averaged over the entire dataset (that is not class-specific) SF from [5]. Here, in addition to observing the **Memorization Transition**, we identify a **Speciation Transition** [6] at time $T_{s}$, where $T_{s}>T_{m}$. When the U-Turn time $T_{u}$ exceeds $T_{s}$, the resulting image is not only distinct from the initial image but also belongs to a different class than the initial image. A schematic illustration of the U-Turn in the case of a multi-class score function is illustrated in Figure 1b.

The next two contributions, in particular the fifth one also discussed in Section 5, came from the follow-up experiments with the averaged-over-the-multi-class score function of the CIFAR-10 dataset. By inspecting synthetic samples, we observe that both $T_{m}$ and $T_{s}$ fluctuate rather significantly from class to class and even from sample to sample. This motivated us to analyze the U-Turn auto-correlation function, however now conditioned to a single GT sample and to a single class. We observe, now quantitatively, that the GT-sample-constrained U-Turn auto-correlation function fluctuates significantly from sample to sample; that is (using statistical physics terminology), not a self-averaged quantity. This lack of self-averageness persists, even though it became weaker, when we compare U-Turn auto-correlation functions (conditioned to a class of the GT sample) for different classes.

Six, we pose in Section 6 the question of whether the dynamic phase transitions at $T_{m}$ and $T_{s}$ are related to the nonlinearity of the SF. To investigate this, we develop a Gaussian Test and a related G-Turn algorithm, assuming the GT data are Gaussian or can be approximated by a Gaussian distribution, which then allows for an analytical expression of the U-Turn auto-correlation function. We observe that, while the empirical U-Turn auto-correlation functions evaluated on the ImageNet dataset and on the averaged-over-all-the-classes CIFAR-10 dataset are relatively close to the corresponding estimate for the U-Turn auto-correlation function computed within the Gaussian ansatz, there is still a notable divergence between them. A detailed examination of the U-Turn sampling procedure in the CIFAR-10 experiments leads to a key final observation: the score function becomes effectively linear for $T_{u}>T_{s}$, and it is approximately linear, however still with notable deviations from linearity detected via the FID test.

We also discuss the empirical results of applying the U-Turn diffusion to the case of a deterministic sampler (reversed process) in Section 7.

We conclude this paper in Section 8 by summarizing the results and sharing some path-forward ideas.

### 1.2. Related Work

In [7], the authors examined diffusion models from the perspective of symmetry breaking, revealing that the early stages of the reverse dynamics contributed minimally to generating new samples. Based on these observations, they proposed a “late start” strategy for the reverse process, which is similar to the U-Turn discussed in this paper. We consider this work and two other papers discussed next as foundational to our more systematic analysis presented here.

In [6], the authors—motivated by the analysis of a toy example where the GT data are sampled from a bimodal distribution represented by a mixture of two Gaussians reported in [8]—reported the emergence of the dynamic phase transitions of the memorization and speciation type. Even though Memorization Transition was reported earlier in the first version of our manuscript, our Speciation Transition analysis on the multi-class model as well as our analytic analysis of the diffusion in the case of the Gaussian GT data reported in this manuscript were inspired by the respective analyses of [6,8].

The phase transition methodology of statistical physics also became central to the “forward-reversed experiments” of [9], where the authors experimented with a setting similar to the U-Turn. They empirically reported a phase transition in the value of $T_{u}$ and provided some statistical physics estimations based on a mean-field approximation.

Another related method, called “boomerang”, was reported in [10]. This method focuses on generating images that closely resemble the original samples, whereas U-Turn diffusion aims to create distinctly different images that approximate i.i.d. samples from the entire dataset.

## 2. Technical Introduction: Score-Based Diffusion

We follow the SBD framework, as expounded in [1]. The SBD harmoniously integrates the principles underlying the “Denoising Diffusion Probabilistic Modeling” framework introduced in [11] and subsequently refined in [3], along with the “Score Matching with Langevin Dynamics” approach introduced by [12]. This seamless integration facilitates the reformulation of the problem using the language of Stochastic Differential Equations, paving the way to harness Anderson’s Theorem [4]. As elucidated in the following, this theorem assumes a principal role in constructing a conduit linking the forward and reverse diffusion processes.

Let us follow [1] and introduce the forward-in-time SDE for the vector $x_{t}$ = $(x_{t;i}|i=1,$$\cdots ,d)\in R^{d}$ (where *d* is the embedding space for the data):(1)$$\begin{array}{cc} \; & \mathrm{Forward}:\;t\in \left[0\to T\right]:\;dx_{t}=f\left(x_{t},t\right)dt+g\left(x_{t},t\right)dw_{t}, \end{array}$$
and another reverse-in-time SDE:
(2)$$\begin{array}{cc} \mathrm{Reversed}:\;t\in \left[T\to 0\right]:\;dy_{t} & =\left(f\left(y_{t},t\right)-\;\nabla \cdot G\left(y_{t},t\right)-G\left(y_{t},t\right)s\left(y_{t},t\right)\right)dt\\ & +\;g\left(y_{t},t\right)d{\bar{w}}_{t}, \end{array}$$
where the drift/advection $f:R^{d}\times R\to R^{d}$ and diffusion $g:R^{d}\times R\to R^{d}\times R^{d}$ are sufficiently smooth (Lipschitz functions) and $G\left(y_{t},t\right)=g\left(y_{t},t\right)g{\left(y_{t},t\right)}^{⊤}$. Here, $s\left(x_{t},t\right)≐\nabla_{x_{t}}\log p\left(x_{t},t\right)$ is the so-called score function computed for the marginal probability distribution of the forward process (Equation 1) and utilized in Equation (Equation 2) to drive the reverse process. Both forward and reversed processes are subject to Ito regularization. $w_{t}$ and ${\bar{w}}_{t}$ represent standard Wiener processes for forward and reverse in time, respectively.

The forward diffusion process transforms the initial distribution $p_{\mathrm{data}}\left(\cdot \right)$, represented by samples, into a final distribution $p_{T}\left(\cdot \right)$ at time *T*. The terms $f(x_{t},t)$ and $g(x_{t},t)$ in the forward SODE (Equation 1) are free to choose, but in the SBD approach, they are usually selected in a data-independent manner.

Anderson’s theorem establishes that the forward-in-time process and the reverse-in-time process have the same marginal probability distribution, provided that the reverse process is initiated at t=T with the marginal probability distribution identical to the marginal forward probability distribution evaluated at t=T. Note that Anderson’s theorem [4] actually establishes a stronger equivalence between the forward process (Equation (Equation 1)) and the reverse process (Equation (Equation 2)): not only do their marginal probability distributions coincide, but their transition probability kernels are identical as well. This result is analogous to the detailed balance (DB) condition in equilibrium statistical mechanics—where, for a system at equilibrium, the rate of transitions from state *i* to state *j* (weighted by the equilibrium distribution) equals the reverse rate—but here it is extended to the non-autonomous (time-dependent) setting of our diffusion processes. In contrast, alternative approaches such as the probability flow method of [1] yield a deterministic reverse process. In these cases, although the transition probabilities do not satisfy DB, the marginal distributions of the forward and reverse processes remain equal. The proof of Anderson’s Theorem relies on the equivalence of the Fokker–Planck equations derived for the direct (Equation 1) and inverse (Equation 2) dynamics: (3)$$\begin{array}{cc} \partial_{t}p+\nabla_{i}\left(f_{i}p\right) & \;=\;\;\frac{1}{2}\nabla_{i}\nabla_{j}\left({\left(gg^{T}\right)}_{ij}p\right), \end{array}$$(4)$$\begin{array}{cc} \partial_{t}p+\nabla_{i}\left(f_{i}p\right)-\nabla_{i}\left(p\nabla_{j}{\left(gg^{T}\right)}_{ij}\right)-\nabla_{i}\left(gg^{T}{)}_{ij}s_{j}p\right) & \;=-\frac{1}{2}\nabla_{i}\nabla_{j}\left({\left(gg^{T}\right)}_{ij}p\right), \end{array}$$
where $i,j=1,\cdots ,n_{x}$ and we assume summation over repeated indexes, and we also dropped dependencies on x or y and on *t* in f,g, and s to make the notations lighter.

**Inference**, which involves generating new samples from the distribution represented by the data, entails initializing the reverse process (Equation 2) at a sufficiently large (but practically finite) t=T with a sample drawn from the normal distribution derived in the limit of T→∞, *p* and then running the process reversed in time to reach the desired result at t=0. This operation requires accessing the SF, as indicated in Equation (Equation 2). However, practically obtaining the exact time-dependent SF is challenging. Therefore, we normally resort to approximating it with an NN parameterized by a vector of parameters θ: $s_{\theta }\left(\cdot ,t\right)\approx s\left(\cdot ,t\right)$.

The NN-based approximation of the SF allows us to efficiently compute and utilize gradients with respect to the input data x at different times *t*, which is essential for guiding the reverse process during inference. By leveraging this NN approximation, we can effectively sample from the desired distribution and generate new images which are approximately i.i.d. from a target probability distribution represented by input data. This approach enables us to achieve reliable and accurate inference in complex high-dimensional spaces, where traditional methods may struggle to capture the underlying data distribution effectively.

**Training:** The NN $s_{\theta }\left(\cdot ,t\right)$ can be trained to approximate $s\left(x_{t}\right)=\nabla_{x_{t}}\log p_{t}\left(x_{t}\right)$ using, for example, the weighted Denoising Score Matching (DSM) objective [1]:$$\begin{array}{c} E_{t∼U\left(0,T\right),x_{0}∼p_{0}\left(\cdot \right),x_{t}∼p_{t}\left(\cdot |x_{0}\right)}\left[\frac{\lambda (t)}{2}{∥\nabla_{x_{t}}\log p_{t}\left(x_{t}|x_{0}\right)-s_{\theta }\left(x_{t},t\right)∥}_{2}^{2}\right]. \end{array}$$In the experiments presented in this manuscript, we do not train the models ourselves. Instead, we leverage pre-trained models (score functions) from existing open-source work specifically [5], for class-conditioned ImageNet, which provides a separate score function for each class, and for multi-class CIFAR-10, which uses a single score function for the entire dataset. Our focus is primarily on analyzing and understanding the basic SBD scheme, and subsequently on improving it by proposing the U-Turn scheme and related modifications.

### 2.1. Choice of SBD—Time-Dependent Brownian Diffusion

In this manuscript, we choose to work with the simplest SBD model—the time-dependent Brownian diffusion: the drift term $f(x_{t},t)$ is zero and the diffusion term $g(x_{t},t)$ is space-independent but time-dependent in—$\sqrt{2\beta_{t}}$. This results, according to Equation (Equation 1), in the following explicit expression for the exact SF:(5)$$\begin{array}{c} s\left(x_{t},t\right)=\nabla_{x}\log\left(\sum_{n=1}^{N}N\left(x_{t}|x^{\left(n\right)};2\hat{I}\int_{0}^{t}dt^{\prime }\beta_{t^{\prime }}\right)\right), \end{array}$$
where n=1,⋯,N indexes the GT samples; $\hat{I}$ is the identity matrix; and $N\left(x|\mu ;\hat{\Sigma }\right)$ is the normal distribution of x with the mean vector μ and covariance matrix $\hat{\Sigma }$.

Two key remarks are warranted regarding the universality of the SBD approach and the selection of the SBD model.

First, we highlight that explicit expressions for the SF, such as the one presented in Equation (Equation 5), are a common feature across various SBD models. This explicit form of the SF eliminates the need to simulate the forward process, offering a significant computational advantage.

Second, an important observation in the field is that, while the details of the forward model are critical for practical implementation, the choice of the underlying model is surprisingly flexible. For instance, the basic model in Equation (Equation 5), which we employ in our experiments, reliably supports the generation of high-quality images across diverse settings.

We now turn to a discussion of the discretization and the selection of the β-protocol, addressing each in the context of the two pre-trained use cases developed in [5].

#### Pre-Trained ImageNet-64 and CIFAR-10

For the ImageNet-64 dataset, https://www.image-net.org/ (accessed on 1 August 2023), the total duration of the SBD process was set to T=80, with the time-dependent diffusion coefficient defined as β(t)=2t. The interval [0,T] was discretized into 255 non-uniform time steps, denoted as $\left\{t_{i}\right\}$. Larger time steps were allocated during the early stages of the reversed SDE and progressively smaller steps in the later stages. Further details on this discretization can be found in Appendix D.1 of [5]. In this setup, the time dependence $t=t_{i}$ is indexed as i=0,⋯,255.

We utilized the open-source data and code from [5], which eliminated the need to retrain the model to generate $s_{\theta }\left(x_{t};t\right)$, an NN approximation of Equation (Equation 5). This significantly reduced computational overhead. The pre-trained model achieved an FID score of 1.36 using 511 Neural Function Evaluations (NFEs), corresponding to evaluating the NN version of Equation (Equation 5) twice per discretization step. In our implementation, we observed a slightly higher FID score of 1.42 for 50,000 generated images. Although this represents a marginal difference, it highlights the robustness of the original method and the reproducibility of the results reported in [5].

For the CIFAR-10 dataset, https://www.cs.toronto.edu/kriz/cifar.html (accessed on 10 August 2023), a similar SBD process was applied with T=80 and β(t)=2t. The time interval [0,T] was discretized into 512 non-uniform steps $\{t_{i}\},i=1,\cdots ,512$, with larger steps in the early stages of the reversed SDE and smaller steps in the later stages, following the approach of [5]. The pre-trained model achieved an FID score of 3.5 with 1024 NFE, where each discretization step involved evaluating the NN version of Equation (Equation 5) twice. In our implementation, the FID score was slightly higher at 3.65 for 50,000 generated images. This small deviation again underscores the robustness of the original approach and the reproducibility of the results.

## 3. Analysis of Basic SBD Model

In the following, and as custom in the field, we will extensively use the FID metric to evaluate the quality of images generated by basic diffusion models and their U-Turn counterparts. The FID measures the similarity between the distributions of GT and generated images by approximating them as normal distributions (see Section A.1 for details). Although the FID score does not assess whether the generated images are correlated with specific GT images, this approach is justified in the context of standard SBD modeling. In such models, the reverse process is initialized independently and identically distributed (i.i.d.) from a normal distribution, preventing the memorization of GT samples.

However, this logic does not apply to our task of analyzing how the ensemble of initial GT images, or a particular initial GT image, becomes forgotten over time as we advance with the forward SDE. To address this limitation in the analysis of SBD models, we introduce additional tests alongside the FID test: the Kolmogorov–Smirnov (KS) and norm of SF test. These tests are discussed and then applied in the following subsections to analyze the SBD models over 1000 generated samples of ImageNet, which serves as our first and class-specific/fixed working example. Later in the manuscript, we will also discuss applications of the tests to the multi-class CIFAR-10 data.

### 3.1. Kolmogorov–Smirnov Test

We employ the Kolmogorov–Smirnov (KS) Gaussianity test to evaluate the null hypothesis: “Is a single-variable marginal of the resulting multivariate distribution of x(t) at a given time *t* Gaussian?” To address this hypothesis, the KS test is applied to each single-variable marginal, defined as $p_{t}\left(x_{k}\right)=\int d\left(x\setminus x_{k}\right)p_{t}\left(x\right)$. For implementation details, refer to Section A.2.

The results of the KS analysis for the reverse process in the ImageNet experiment are shown in Figure 2.

We observe a smooth yet significant transition in the KS factor over time, starting around i≈50, where $p_{t}\left(x_{t}\right)$ is far from Gaussian, and progressing to i≈100, where the distribution becomes much closer to Gaussian. This transition suggests that the process of encoding information from the GT samples into the time- and GT-sample-dependent score function is largely completed by *i* ≈ 100–200.

Interestingly, the initial level of non-Gaussianity differs across labels at i=0. However, as time progresses, the KS curves for different labels converge, becoming nearly identical. This indicates a uniform Gaussianization process across labels as the reverse process evolves.

Additionally, further “Gaussianization” is observed at i≈225, as evidenced by a decrease in the KS factor. It is important to note that the KS test evaluates only the Gaussianity of spatial marginals—associated with a single component of x. This restricted Gaussianity does not necessarily imply that the entire vector x follows a Gaussian distribution.

We also explored various bivariate KS tests, but these exhibited significant variability across tests. This variability motivates the exploration of additional quantitative methods, which are discussed in subsequent sections. The effective Gaussianity of the forward process at later times and the reversed process at early times will be revisited in Section 6 and Appendix B.

### 3.2. Average of the Normalized Score Function 2-Norm

The time dependence of the normalized 2-norm of the score function, averaged over multiple paths or instances of the reversed process, is presented in Figure 3. Details of this computation can be found in Section A.3.

Consistent with the dynamics observed in the Kolmogorov–Smirnov (KS) analysis (Figure 2), significant changes in the normalized score function 2-norm begin at approximately *i* ≈ 50 and are largely completed by *i* ≈ 100–150. This behavior suggests that, if the score function norm is used as a criterion, no substantial additional information about the GT samples is being encoded into the score function at later times.

These observations align with the trends and discussions presented in subsequent sections, further supporting the interpretation of different stages in the reverse process dynamics.

### 3.3. Insensitivity to Reverse Process Initialization

In standard SBD modeling, the reverse process is typically initialized with an i.i.d. sample drawn from the probability distribution of the forward process at time *T*, or at T→∞ if such a limit exists. This distribution is often a simple Gaussian. However, as demonstrated in Figure 4, the specifics of the reverse process initialization appear to have minimal impact on the generated output samples. The figure shows results obtained from the same model but initialized with different distributions.

This insensitivity to the specifics of reverse process initialization, combined with the early temporal saturation observed in both the KS and score function norm tests discussed in previous subsections, suggests the potential to shorten the durations of both the forward and reverse processes. Motivated by this observation, we shift our focus to exploring this approach in the next section.

Bridging Theory and Empirical Analysis: The theoretical framework developed in this and preceding sections rigorously characterizes the dynamics of the forward and reverse diffusion processes, highlighting key properties such as detailed balance and the evolution of the score function. To connect these theoretical insights with practical applications, the subsequent sections focus on empirical validation through image synthesis. In the next section, we introduce the U-Turn diffusion algorithm, which operationalizes the theoretical principles to improve computational efficiency and image quality. This transition not only demonstrates how abstract diffusion dynamics inform the design of our algorithm but also provides a clear pathway from mathematical analysis to tangible image analysis outcomes.

## 4. U-Turn

Motivated by the analysis in the preceding section, we introduce a modification of the basic SBD process, which we call U-Turn. Our analysis of the dynamics of the basic SBD process, particularly when run for a sufficiently long time (i=255 time steps in the ImageNet experiments), suggests that such an extended duration may not be necessary to generate high-quality images. Instead, we propose making a U-Turn: identifying an appropriate time $T_{u}$ (and equivalently discrete index $i_{u}<255$) and then initializing the reverse process using the SF as in Equation (Equation 2), or its NN approximation, at this shorter time. The initialization involves a sample generated from the explicitly known distribution $p_{t}\left(\cdot |x_{0}\right)=N\left(\cdot |0;\beta \left(t\right)\hat{I}\right)$ that is conditioned to a particular choice of a GT sample for $x_{0}$. See Algorithm 1.
**Algorithm 1:** U-Turn**Require:** $s_{\theta }\left(x_{t},t\right)$ – NN approximation of $s_{\theta }\left(x_{t},t\right)$ defined by Equation (Equation 5); $x_{0}∼p_{\mathrm{GT}}$, $T_{u}$.
 Initialize the reversed process: $y_{T_{u}}∼N\left(\cdot |x_{0};\beta \left(T_{u}\right)\hat{I}\right)$. Run the reversed process according to Equation (Equation 2). Output the newly generated/synthetic image $y_{0}$.


The algorithm depends on $T_{u}$, raising the significant question: how do we identify an optimal, or simply appropriate $T_{u}$? One option, suggested by the analysis in Section 3, is to introduce criteria based on one of the tests—a KS test or SF-norm test. For example, we can choose $T_{u}$ by setting the SF-norm such that its derivative reaches a predefined small value, indicating that the SF-norm stops changing.

Alternatively, we can experiment by scanning different values of $T_{u}$. We can start by relying on both our visual perception and FID tests of the U-Turn.

### 4.1. Visual Examination on the ImageNet Model

Images generated using Algorithm 1 applied to the ImageNet dataset are displayed in Figure 5. These figures illustrate the evolution of output images in the U-Turn scheme as a function of $T_{u}$, conditioned on the class (note that the class remains unchanged from the GT sample through to the initialization of the reverse process after the U-Turn).

A visual inspection reveals a transition around $i_{u}\in \left[150,200\right]$, where the model shifts from reproducing the initial images observed at earlier times to generating new images from the same class at later times. We term this dynamic (phase) transition the **Memorization Transition** and denote the corresponding transition time as $T_{m}$. Furthermore, within experiments displayed in the same column of Figure 5—each using a different $T_{u}$ but initialized with the same image and random seed for the reverse process—the transition appears relatively sharp. In other words, we consistently observe clear images rather than artifacts (e.g., noisy images or mixed representations). Notably, for $T_{u}>T_{m}$, the generated images vary with different values of $T_{u}$ within the same column.

Moreover, moving from one column to the next—which corresponds to different initial images and labels—reveals some significant variation in the specific time at which the transition $T_{m}$ occurs; that is, dependence on the GT sample and its class—see Section 5.2 for further discussion of the strong sensitivity to the GT sample.

In summary, based on our visual analysis of Figure 5, we conclude that to generate new images, a U-Turn at $T_{u}>T_{m}$ is required. The value of $T_{m}$ appears to depend on both the class and the underlying SBD model.

### 4.2. FID Test: U-Turn vs. Artificial Initialization

Let us analyze the U-Turn quantitatively using the FID score. However, in addition to running the U-Turn algorithm in its basic form, as shown in Algorithm 1, we will experiment with replacing the U-Turn-specific initialization of the reverse process with $y_{T_{u}}$ generated artificially—independently of the GT sample used in Algorithm 1. The results of these experiments are shown in Figure 6. Several useful observations can be deduced from these results.

First, we observe that a low FID score for the U-Turn alone is not indicative of the diversity of the generated images. When the U-Turn is made at a sufficiently small $T_{u}$, a low FID simply corresponds to the memorization of the initial GT image. Conversely, when we test the FID of the U-Turn jointly with the initialization of the reverse process at the same $T_{u}$, but with a sample independent of the GT image, these combined results indicate whether the respective $T_{m}$ has been reached. Specifically, if the two processes show comparable FID scores, then $T_{u}$ is optimal or close to optimal. We also see that if $T_{u}\geq T_{m}$, a number of alternative initializations become comparable to the U-Turn in terms of their FID performance. Finally, all artificial initializations result in ridiculously large FID scores if $T_{u}$ is too short.

We conclude that the “U-Turn vs. artificial initialization” comparison provides an explicit method for finding the optimal U-Turn time, $T_{m}$. For the ImageNet case, this suggests that $i_{m}\approx 200$ (the index of $T_{m}$).

### 4.3. Auto-Correlation Function of U-Turn

The approach described in the previous subsection, Section 4.2, is strong as it allows for a direct test of the U-Turn performance. This contrasts with the implicit suggestions for selecting $T_{u}$ based on KS and SF-norm tests, which track the temporal evolution within the basic SBD algorithm. However, one limitation of the “U-Turn vs Artificial Initialization” test in Section 4.2 is its empirical nature. We aim to design a test that explicitly refers to the U-Turn and relies on a measure of correlation loss within the U-Turn process between the GT sample and the resulting image. The following construction meets this requirement.

Here, we test the “independence” of the newly generated synthetic samples from their respective GT samples for U-Turns at different $T_{u}$. Specifically, we compute the U-Turn auto-correlation (AC) function:(6)$$\begin{array}{c} C_{UT}\left(T_{u}\right)=\frac{1}{N}\sum_{n=1}^{N}\frac{{\left(x^{\left(n\right)}\left(0\right)\right)}^{T}y^{\left(n\right)}\left(0\right)}{{\left(x^{\left(n\right)}\left(0\right)\right)}^{2}}, \end{array}$$
where $x^{\left(n\right)}\left(0\right)$ is the *n*-th sample from the GT, resulting in $x^{\left(n\right)}\left(T_{u}\right)∼P_{\mathrm{forw}.\;\mathrm{proc}.}\left(\cdot |T_{u};x^{\left(n\right)}\left(0\right)\right)$ generated at $t=T_{u}$ and then used to initialize the reversed process at the same time, producing a sample path $y^{\left(n\right)}\left(T_{u}\to 0\right)$ that arrives at t=0 as $y^{\left(n\right)}\left(0\right)$. This type of AC function will decay with $T_{u}$, allowing us to directly quantify when to stop based on a sufficiently small value, which may be dependent on the class.

The results for $C_{UT}\left(T_{u}\right)$ are presented in Figure 7, which shows a monotonic decrease in $C_{UT}\left(T_{u}\right)$ with increasing $T_{u}$ ($i_{u}$) across all classes. We observe that the curves eventually saturate, though at values that vary considerably across classes. We attribute this phenomenon to certain distinctive features that may be relatively uniform within a given class—for example, the predominance of orange hues within the “oranges (fruits)” class—which can sustain higher U-Turn auto-correlations within the class even at a large $T_{u}$.

This observation suggests that the most reliable approach for determining $T_{m}$ might involve setting a threshold based on visual perceptions of image quality. However, this threshold should be adjusted according to the cross-correlation values within the class, specifically the value of the U-Turn auto-correlation function at the maximum time, $C_{UT}\left(T=255\right)$. Based on this, as well as our visual assessment (see Figure 5), we propose a practical criterion by setting $1.2\times C_{UT}\left(T=255\right)$ as the threshold, i.e., $C_{UT}\left(T_{m}\right)=C_{UT}\left(T=255\right)$. For our test case with ImageNet, this threshold corresponds to approximately $T_{m}=i_{m}\approx 200$, though variations are observed across different classes.

#### 4.3.1. Gaussian Approximation for the U-Turn Auto-Correlation Function

If the entire GT dataset were Gaussian, the U-Turn auto-correlation (AC) function could be computed analytically, as both the forward and reverse processes would also be Gaussian in this case. The respective computations are detailed in Appendix B. Using this Gaussian theory, we approximate the AC function for the GT dataset by computing its covariance while ignoring non-Gaussian contributions.

A comparison between Gaussian theory and the empirical evaluation of the U-Turn AC function for the entire ImageNet dataset (averaged across all classes) is shown in Figure 8. Notably, we observe a reasonably good fit between Gaussian theory and the empirically averaged curve. The close agreement between the two suggests that a global “annealed” characterization of the U-Turn behavior is Gaussian, effectively smoothing out sample-specific and class-specific variations which we saw above (in the main part of this subsection) analyzing the U-Turn AC function conditioned to classes.

## 5. Multi-Class—The Case of CIFAR-10

We now turn to discussing how the U-Turn approach, along with the performance tests introduced and analyzed thus far, performs in the case of an unsupervised multi-class setting; that is, when a single score function is used to train on datasets containing samples from multiple classes without accounting for the class label of each sample. We begin with a visual inspection of the U-Turn tested on the pre-trained multi-class but single SF CIFAR-10 model.

### 5.1. Visual Inspection and FID

Since our primary motivation for experimenting with CIFAR-10 is to gain a better understanding of the multi-class nature of the dataset and its representation within a single (cumulative) SF, we begin our visual analysis of Figure 9 by noting that, at sufficiently large values of $T_{u}$, the U-Turn process starts generating images from different classes. This observation aligns with the predictions of [6] regarding the emergence of the **Speciation Transition**, it is worth noting, however, that the findings of [6] pertain to the standard SBD setting, not specifically our U-Turn modification. Therefore, a more accurate statement would be that we observe a generalization of the Speciation Transition predicted in [6] to the U-Turn setting. We estimate that, depending on the column (each representing experiments with the same initial image and progressively increasing values of $T_{u}$ from top to bottom), the transition occurs in the range $T_{s}\in \left[250,400\right]$.

Interestingly, in some cases, images from new classes appear early on, but at larger values of $T_{u}$, the original class can re-emerge. We attribute this to the fact that the probability of the initial state of the reverse process falling within the domain associated with the same label as the GT sample remains non-zero.

Next, we observe that memorization of the initial image ceases to be an issue at somewhat earlier times, which we naturally identify as the **Memorization Transition**, $T_{m}$, noting that $T_{m}<T_{s}$. As in the previously discussed case with a single-class SF, the values of $T_{m}$ vary from column to column (from one initial GT sample to another) and are observed within the range $T_{m}\in \left[150,250\right]$.

Turning our attention to image quality (which, as seen in Figure 5, remained consistently high across different $T_{u}$ values in our fixed-class ImageNet experiments), we find that in the multi-class case, image quality varies. In the memorization phase, when $T_{u}$ is relatively small, image quality is high—we consistently obtain outputs that are clear and legitimate images. However, as we progress to larger $T_{u}$ values, beyond the Memorization Transition but still before the Speciation Transition $T_{s}$, we start to observe images within individual sequences (columns in Figure 9) that are less clear. Eventually, for sufficiently large values of $T_{u}>T_{s}$, the quality of the output images improves again.

The non-monotonic dependence on $T_{u}$ of the image quality is also seen quantitatively in Figure 10, where we show the FID score computed for CIFAR-10 as a function of $T_{u}$ in the U-Turn process.

In summary, our visual examination of Figure 9, complemented by the quantitative analysis of the FID in Figure 10, suggests a dynamic phase transition structure within the U-Turn scheme that generally aligns with the observations of [6] regarding dynamic phase transitions in the standard SBD model: we identify a Memorization Transition at $T_{m}$ and a Speciation Transition at $T_{s}$, with $T_{m}<T_{s}$. However, we also report new findings specific to the U-Turn approach:Both $T_{m}$ and $T_{s}$ vary depending on the initial GT sample.The region between $T_{m}$ and $T_{s}$ appears somewhat blurry, with the potential for some newly generated samples to be noisy or ambiguous (a mix of images).There may be more than one Speciation Transition—possibly a hierarchy of transitions, each corresponding to diffusion from the domain of the initial GT sample to a domain associated with a different class.

### 5.2. Conditional U-Turn: Quantitative Analysis

The concluding remark of the preceding subsection, based on the visual inspection of the newly generated images, should be treated as a hypothesis. Here, we aim to validate this hypothesis through a quantitative analysis, which involves evaluating the results using the U-Turn auto-correlation function introduced in Section 4.3.

Specifically, we select a particular GT image and initialize the U-Turn process using the same image. The resulting trajectories are recorded, and the conditional U-Turn auto-correlation function is computed using Equation (Equation 6) with N=1, corresponding to the selected GT sample. This procedure is repeated for a set of GT samples drawn from various classes, with representative results presented in Figure 11.

Our analysis of the conditional U-Turn auto-correlations reveals strong dependencies, not only on the class of the initial GT image but also on the specific GT image itself. For example, Figure 11 demonstrates that U-Turn auto-correlations conditioned on different samples—two GT samples from the “plane” class (left panel) and two GT samples from the “trucks” class (right panel)—exhibit noticeable differences both across classes and within samples of the same class.

In addition, Figure 11 compares these conditional U-Turn auto-correlations with their empirical class-averaged counterparts. The class-averaged curves are obtained by averaging the U-Turn auto-correlation functions over a large number of initial GT samples (1000 samples per class). This comparison highlights the variability induced by individual samples when compared to the smoother, class-averaged behavior.

Furthermore, we compute the U-Turn correlations under the assumption that the process is Gaussian, utilizing the formulas derived in Appendix B. This Gaussian approximation relies primarily on the covariance structure of the GT samples within each class. However, as shown in Figure 11, the Gaussian approximation fails to closely match the empirical class-averaged results, indicating that the reverse process introduces significant non-Gaussianity. This discrepancy underscores the complexity of the reverse dynamics, which cannot be captured through simple Gaussian assumptions.

In summary, the conditional U-Turn analysis underscores the dual influences of individual GT samples and class-level features on the resulting correlations. Moreover, it reveals the substantial nonlinearity (and thus non-Gaussianity) of the reverse processes.

### 5.3. From Sample-Conditional to Averaged U-Turn

We now extend the analysis of the U-Turn auto-correlation function, transitioning from the sample-conditional and class-conditional results to a fully averaged perspective over the CIFAR-10 dataset. In this case, we compute the U-Turn auto-correlation function empirically by averaging over all GT samples and across all classes within CIFAR-10. The corresponding results are shown in Figure 12.

In this figure, we compare the fully averaged empirical U-Turn auto-correlation function with its Gaussian counterpart derived under the assumptions outlined in Appendix B. Unlike the results observed for sample-specific and class-specific conditioning, where significant deviations and non-Gaussian effects were evident, the fully averaged U-Turn auto-correlation function exhibits a good agreement with the Gaussian approximation.

Recall that we previously reported the results of the U-Turn AC function analysis, averaged over classes, for the ImageNet dataset in Section 4.3.1. We now demonstrate that the effective Gaussianity of the class-averaged U-Turn AC function is not specific to the ImageNet dataset but is, in fact, a universal phenomenon, also observed in CIFAR-10.

This observation reveals that while the U-Turn auto-correlation function remains a highly informative object for characterizing the forward and reverse processes, it is not a self-averaged quantity in the statistical physics sense. Specifically, our earlier analysis demonstrated considerable variability, nonlinearity, and non-Gaussianity in the conditional U-Turn correlations. However, once averaged across all samples and classes (annealed averaging), these complexities are effectively smoothed out (lost), and the U-Turn behavior becomes almost indistinguishable from that predicted by a Gaussian approximation.

## 6. Gaussian Analysis and Gaussian- (G-) Turn

Appendix B introduces a Gaussian analysis that models GT data as Gaussian, extracting the covariance matrix ${\hat{\Sigma }}_{0}$ and computing U-Turn correlations under this assumption. This section extends the analysis to explore the nonlinearity and non-Gaussianity in empirical processes.

As discussed in Section 5.2, an analysis of the conditional U-Turn on CIFAR-10 data reveals that the reversed process is empirically nonlinear and non-Gaussian. Despite this, it is also natural to conjecture that linearity and Gaussianity emerge in the reverse process at least at sufficiently large times *t* ($T_{u}>t>T_{s}$) but may also be approximately correct at the moderate values of $t\in [T_{m},T_{s}]$ starting from the Memorization Time $T_{m}$. This means that the U-Turn performed at $T_{u}>T_{s}$ and event at $T_{u}\in \left[T_{m},Ts\right]$ would therefore render the early stages of the reversed process approximately linear and Gaussian.

Motivated by these insights, we propose the G-Turn procedure, described in Algorithm 2.
**Algorithm 2:** G-Turn**Require:** $s_{\theta }\left(x_{t},t\right)$: NN approximation of $s(x_{t},t)$ from Equation (Equation 5); ${\hat{\Sigma }}_{0}$.
Initialize a hypothetical reversed process at $T>T_{g}$ (ideally T=∞) with $y_{T}∼N\left(\cdot |0,I\right)$.Compute (not simulate) $y_{T_{g}}$ using the fully Gaussian procedure described in Appendix B, treating the forward process as Gaussian at $t=T_{g}$ with respective mean and variance. Use the time-shifted version of Equation (Equation 12) for this computation.Simulate $y(t\in \left[T_{g}\to 0\right])$ according to Equation (Equation 2), initializing at $t=T_{g}$ with the result from the previous step. This simulation follows the empirical (nonlinear) score function.Output the generated synthetic image $y_{0}$.


We applied Algorithm 2 to the unlabeled CIFAR-10 data, and the results are shown in Figure 13 and Figure 14. A visual examination of Figure 13 reveals that the conjecture described above indeed holds true—the G-Turn becomes successful only at sufficiently large values of $T_{g}$, which we estimate to be compatible with the Memorization Time, $T_{s}$. However, the quantitative analysis depicted in Figure 14 also suggests that the FID score of the G-Turn algorithm reaches the lowest value (correspondent to the performance of the original SBD algorithm) only when $T_{g}\approx T_{s}$. The FID performance is worse (degrades) while still remaining to be reasonably satisfactory when $T_{g}\in \left[T_{m},T_{s}\right]$.

Note that Algorithm 2 may be compared to the approach described in [7], which initializes the reversed process at $T_{g}$ with a Gaussian sample where the mean and variance are computed empirically. In light of our discussion of DB in Section 2 (see the footnote), this approach is effectively equivalent to the G-Turn procedure described here.

Also, a related discussion in [13] addresses the linearization (or Gaussianization) of diffusion samplers. The authors analyze a deterministic reversed process (lacking stochastic components) and develop an analytical model assuming a Gaussian distribution with an arbitrary covariance matrix, represented via singular value decomposition to allow rank deficiency. They provide analytical expressions for the score function and propose approximations based on applying this model to singular value decompositions of actual GT data. A key empirical finding, consistent with ours, is that the linear score function model (the Gaussian approximation of the marginal distribution) holds at large times in the forward process (and early times in the reversed process) but fails during the earliest stages of forward evolution.

## 7. U-Turn for Deterministic Samplers

Although the U-Turn is most naturally applied when the diffusion in the reverse process matches that of the forward process—thus ensuring detailed balance, as extensively discussed above—it can also be utilized in scenarios where DB is broken. Specifically, the U-Turn can be applied to deterministic samplers. In this case, the right-hand side in Equation (Equation 4) is replaced by zero, the score function-dependent term on the left-hand side of the equation is multiplied by 1/2, and respective corrections are made in Equation (Equation 2).

We illustrate the performance of the U-Turn for deterministic samplers in Figure 15 and Figure 16, applied, respectively, to models trained unconditionally (across multiple classes) on CIFAR-10 data and Flickr-Faces-HQ (FFHQ) data, https://github.com/NVlabs/ffhq-dataset (accessed on 10 August 2023).

Several observations can be made based on these results:Lack of randomness in the output images: Examining the deterministic sampler for CIFAR-10 (Figure 15) and FFHQ (Figure 16), we observe that when the U-Turn occurs at $T_{u}>T_{m}$, there is no randomness (uncertainty) in the output image. For deterministic reverse processes, the resulting image remains unchanged with further increases in $T_{u}$ (at least for a sufficiently large $T_{u}$). This is in sharp contrast to stochastic reverse processes, which are the primary focus of this paper, where images generated for different values of $T_{u}$ exhibit noticeable variation.Absence of Speciation Transitions in some cases: In some instances, no Speciation Transition is observed (e.g., in the case of a car, as shown in the third column of Figure 15). This suggests that Speciation Transitions may not always manifest in individual dynamics but could emerge in some form of averaged behavior.Intermediate changes: in several cases (though not all), there is a range of U-Turn times during which changes in the output image are observed, despite the reverse dynamics being deterministic.Class changes with increasing U-Turn time: in some instances, we observe that classes change multiple times as $T_{u}$ increases, potentially due to spontaneous and dynamic symmetry breaking.

These observations highlight unique aspects of applying the U-Turn to deterministic samplers and contrast them with the stochastic processes discussed earlier in this paper.

## 8. Conclusions and Future Work

In this work, we introduced the U-Turn diffusion model as an efficient augmentation of pre-trained score-based diffusion models. Our analysis revealed that by strategically shortening the forward and reverse processes, the U-Turn method retains essential information from the ground truth samples while generating high-quality synthetic images. In particular, we identified two critical time scales: the Memorization Time ($T_{m}$) and the Speciation Time ($T_{s}$), which mark distinct phases in the reverse process dynamics.

The proposed method offers several concrete benefits:**Computational efficiency:** by reducing the duration of both the forward and reverse diffusion processes, the U-Turn model significantly lowers computational overhead while maintaining high image quality.**Controlled diversity:** The U-Turn framework enables a controlled transition from memorization to speciation. This mechanism allows the generation of diverse synthetic images that preserve key features of the ground truth, thereby enhancing the overall utility of the generated data.**Integration with feature-based albumentation:** Our method can be readily integrated into feature-based albumentation pipelines. By increasing the diversity and quality of augmented datasets, the U-Turn model has the potential to improve downstream tasks such as image classification and object detection.**Robustness across settings:** extensive experiments on datasets such as ImageNet and CIFAR-10 demonstrate that the U-Turn approach consistently improves performance across different settings and model architectures.

It is important to note that our analytical insights are derived under the Gaussian assumption, which, while illustrative, does not fully capture the complexities of real-world non-Gaussian dynamics; addressing this limitation is a key direction for future research.

Our experimental results indicate that the U-Turn diffusion model not only accelerates the sampling process but also robustly maintains the essential characteristics of the original data, leading to enhanced synthetic data quality. Future work will focus on optimizing the integration of the U-Turn method within advanced image augmentation frameworks and exploring its application to a broader range of generative tasks.

We believe that these findings underscore the practical advantages of the U-Turn diffusion model and provide a more comprehensive understanding of the trade-offs involved in score-based diffusion processes.

## Figures and Tables

**Figure 1 entropy-27-00343-f001:**
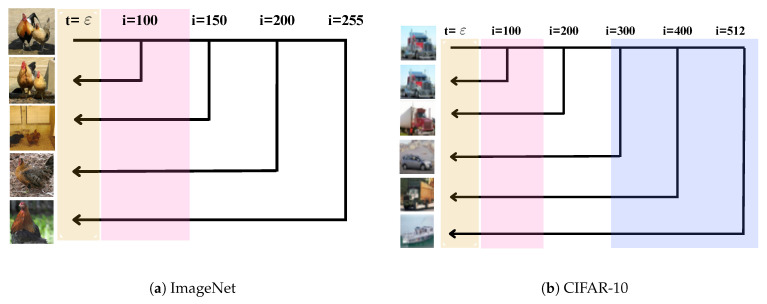
Illustration of the U-Turn concept for (**a**) ImageNet and (**b**) CIFAR-10. Our analysis, based on newly introduced tests, reveals that making the U-Turn earlier is beneficial, but not too early. The dark yellow region indicates a small vicinity near the origin where the approximation of the score function by an NN is crucial to avoid memorization (i.e., generation of GT samples). The pink regions mark the range where Memorization Transitions, $T_{m}$, occur. These transitions are observed in both single-class (ImageNet) and multi-class settings. In the multi-class case (with a single score function for the entire dataset), we also observe the Speciation Transition, $T_{s}$, previously reported in [6] and depicted in light blue in the right figure. This schematic illustration emphasizes an important observation of this work: both $T_{m}$ and $T_{s}$ are not self-averaged. Instead, they fluctuate across GT samples and even between different realizations of the forward process—thus resulting in ranges (distributions).

**Figure 2 entropy-27-00343-f002:**
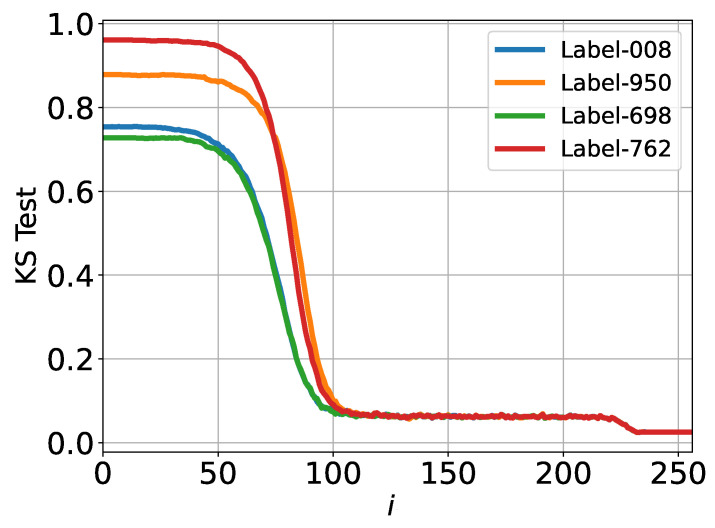
Kolmogorov–Smirnov (KS) test applied to the ImageNet dataset. Different colors represent different classes/labels: 008 (hen), 950 (orange), 698 (palace), and 762 (restaurant, eating house, and eatery).

**Figure 3 entropy-27-00343-f003:**
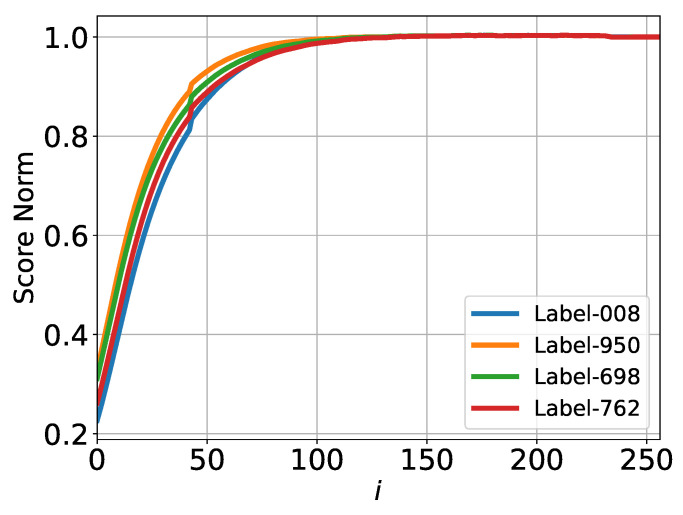
Average of the normalized score function 2-norm test for ImageNet. Consistent with Figure 2, different colors represent different classes/labels.

**Figure 4 entropy-27-00343-f004:**
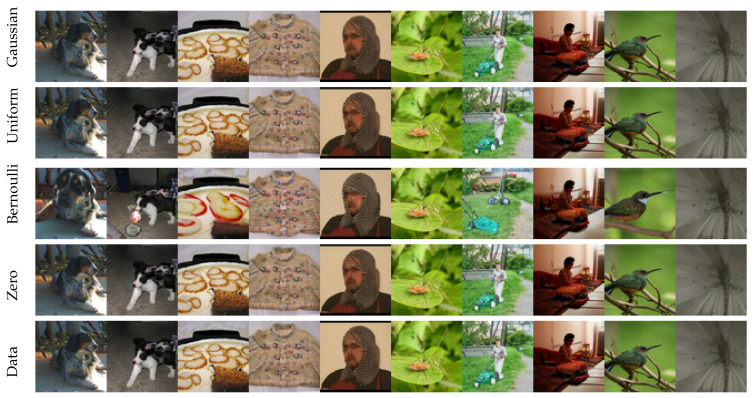
Results of running the reverse process with different initializations: Gaussian ($x_{T}∼N\left(0,I\right)$), uniform ($x_{T}∼\mathrm{Uniform}\left[-1,1\right]$), zero ($x_{T}=0$), GT data ($x_{T}∼p_{\mathrm{data}}$), and Bernoulli ($x_{T}∼p_{\mathrm{Bernoulli}}$).

**Figure 5 entropy-27-00343-f005:**
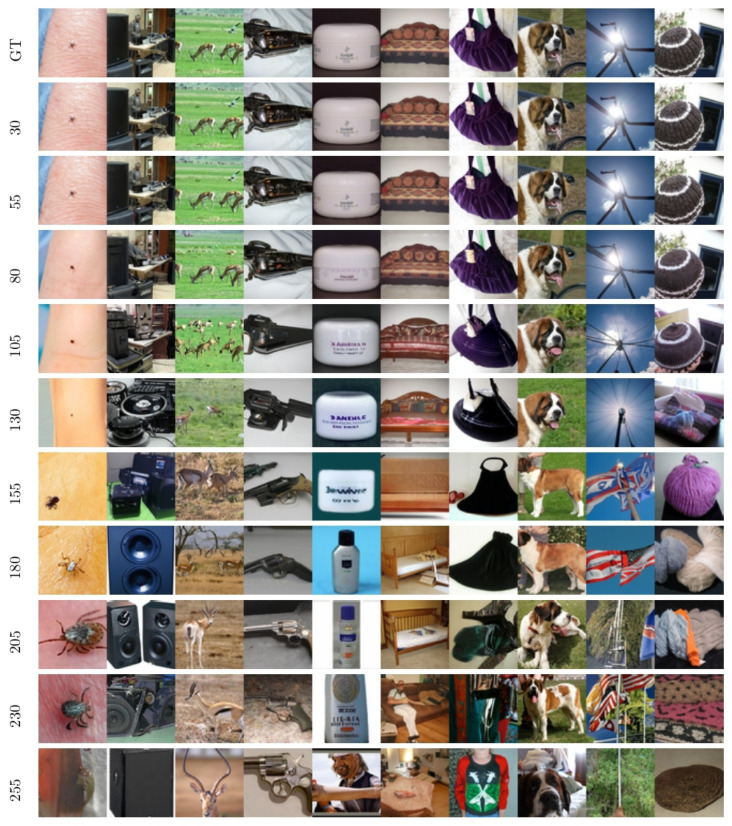
ImageNet visualization: U-Turn at different times $T_{u}$ with inputs of the forward and reverse processes conditioned to the same class.

**Figure 6 entropy-27-00343-f006:**
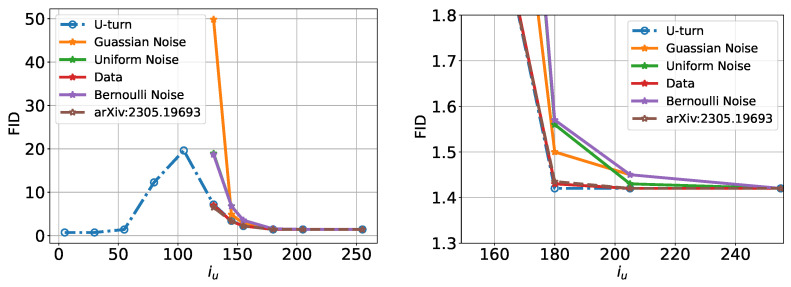
FID score of U-Turn and its variations with different initializations of the reverse process as a function of $i_{u}$ (the discrete index of $T_{u}$). The right sub-figure is a zoomed-in version of the left sub-figure.

**Figure 7 entropy-27-00343-f007:**
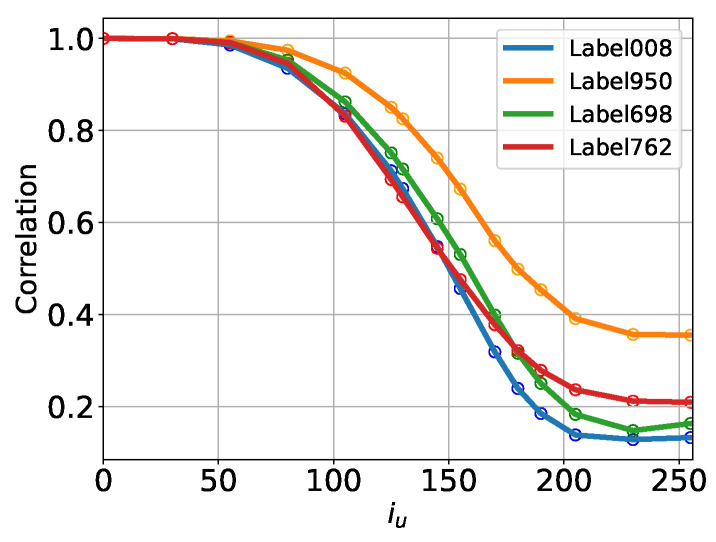
ImageNet: U-Turn auto-correlation functions for $C_{UT}\left(T_{u}\right)$, defined in Equation (Equation 6), conditioned to labels and for each label evaluated over 1000 generated samples. See Figure 2 for description of the classes.

**Figure 8 entropy-27-00343-f008:**
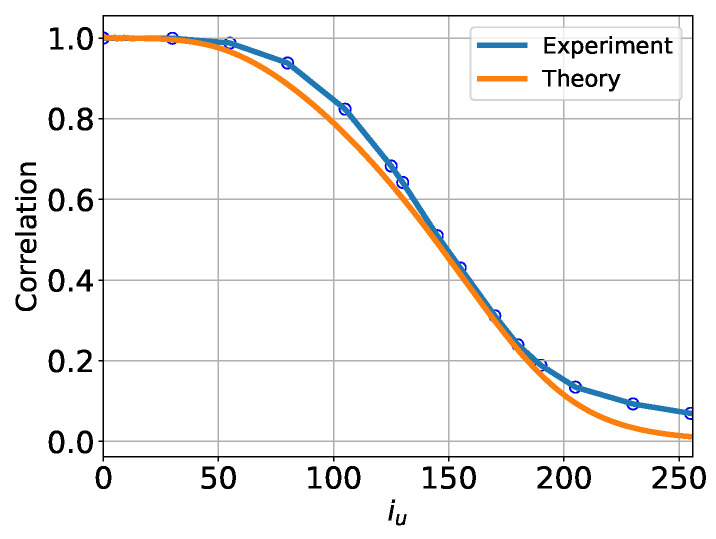
Fully averaged U-Turn auto-correlation functions for the ImageNet dataset, obtained empirically by averaging over all samples and classes, compared to the corresponding fully averaged Gaussian approximation from Appendix B.

**Figure 9 entropy-27-00343-f009:**
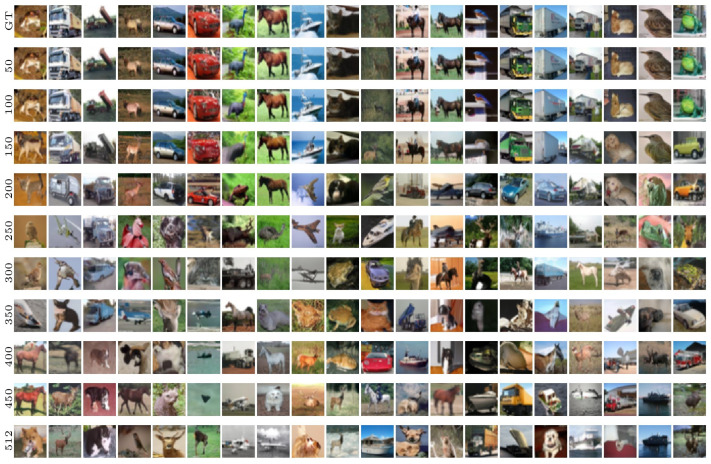
Unconditional CIFAR-10 image generation. Forward dynamics (computations) are variance-exploding (driftless). Reverse dynamics (simulations) are discretized with 512 steps with stochastic samplers.

**Figure 10 entropy-27-00343-f010:**
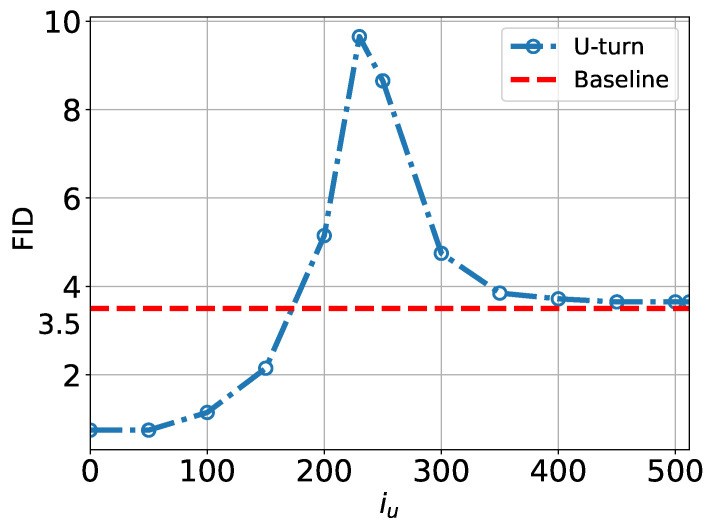
FID score as a function of $T_{u}$ in U-Turn process applied to CIFAR-10 data. Baseline corresponds to the FID score computed over the dataset for the standard SBD.

**Figure 11 entropy-27-00343-f011:**
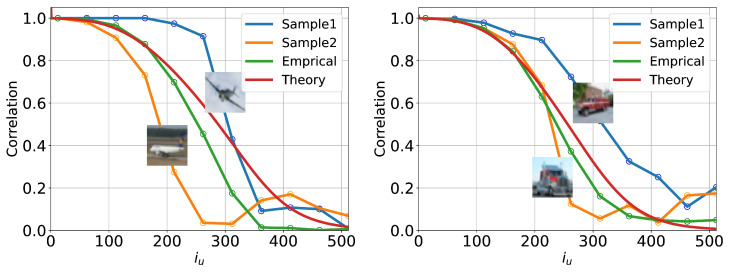
U-Turn auto-correlation functions conditioned on individual GT samples compared to empirical class-averaged results and the Gaussian approximation from Appendix B. (**Left panel**) results for two distinct GT samples from the “plane” class of CIFAR-10. (**Right panel**) results for two distinct GT samples from the “truck” class of CIFAR-10. The analysis highlights the variability in U-Turn correlations across individual samples within the same class, as well as deviations from the Gaussian theoretical prediction.

**Figure 12 entropy-27-00343-f012:**
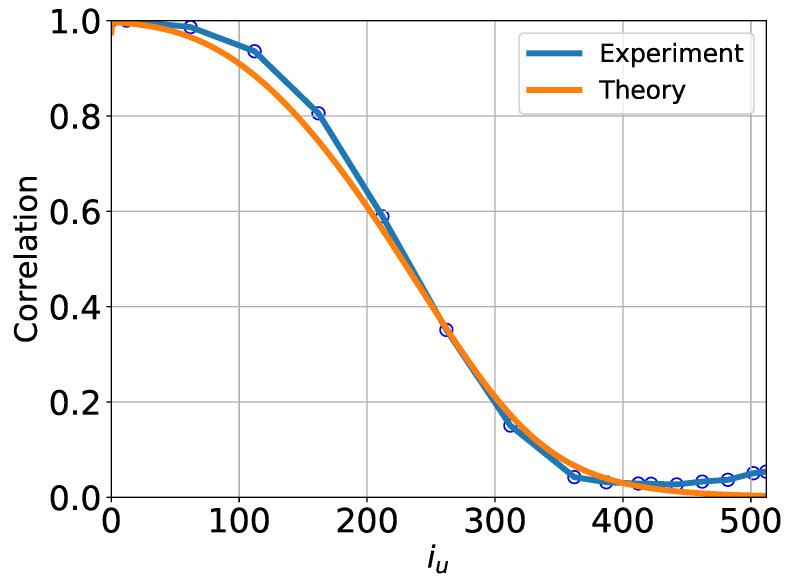
Fully averaged U-Turn auto-correlation functions for the CIFAR-10 dataset, obtained empirically using 5000 samples by averaging over all samples and across all classes, compared to the corresponding fully averaged Gaussian approximation from Appendix B. The close agreement between the two provides a global “annealed” characterization of the U-Turn behavior, effectively smoothing out sample-specific and class-specific variations.

**Figure 13 entropy-27-00343-f013:**
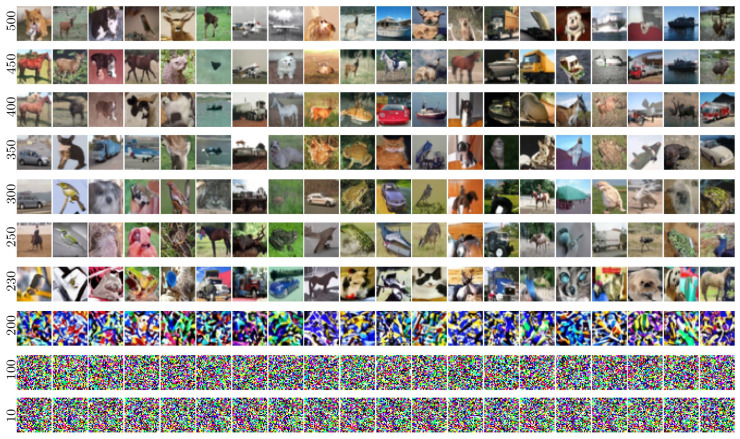
Performance of the G-Turn Algorithm 2 on CIFAR-10. The score function is trained without class conditioning. Columns correspond to different trajectory paths while rows represent different values of $T_{g}$.

**Figure 14 entropy-27-00343-f014:**
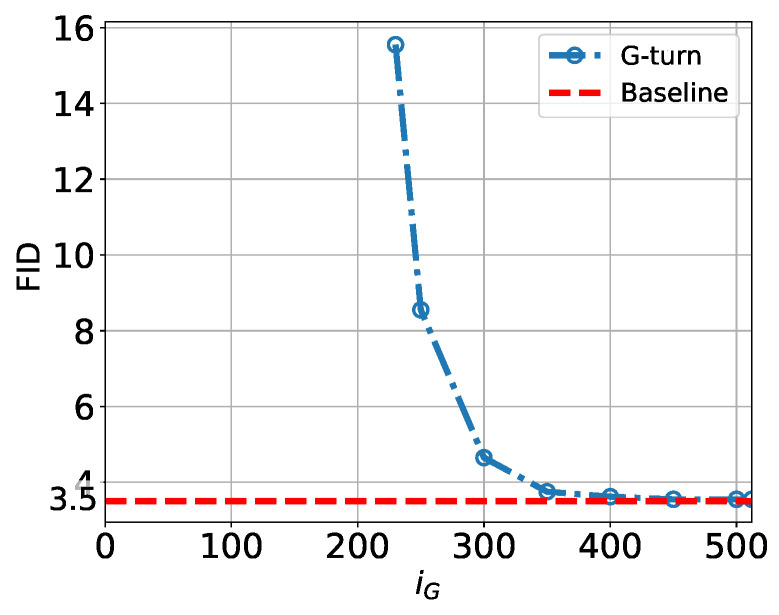
FID scores for the G-Turn Algorithm 2 applied to CIFAR-10, corresponding to the experimental setup visually illustrated in Figure 13.

**Figure 15 entropy-27-00343-f015:**
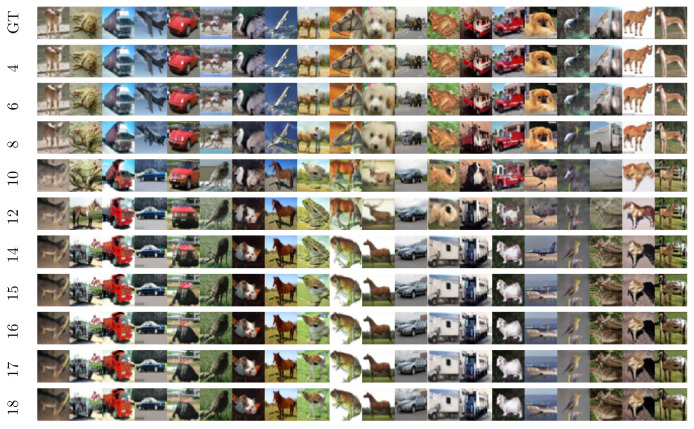
Visual illustration of deterministic sampler for the U-Turn at different times $T_{u}$ over the unconditional (over classes) CIFAR-10 dataset [5].

**Figure 16 entropy-27-00343-f016:**
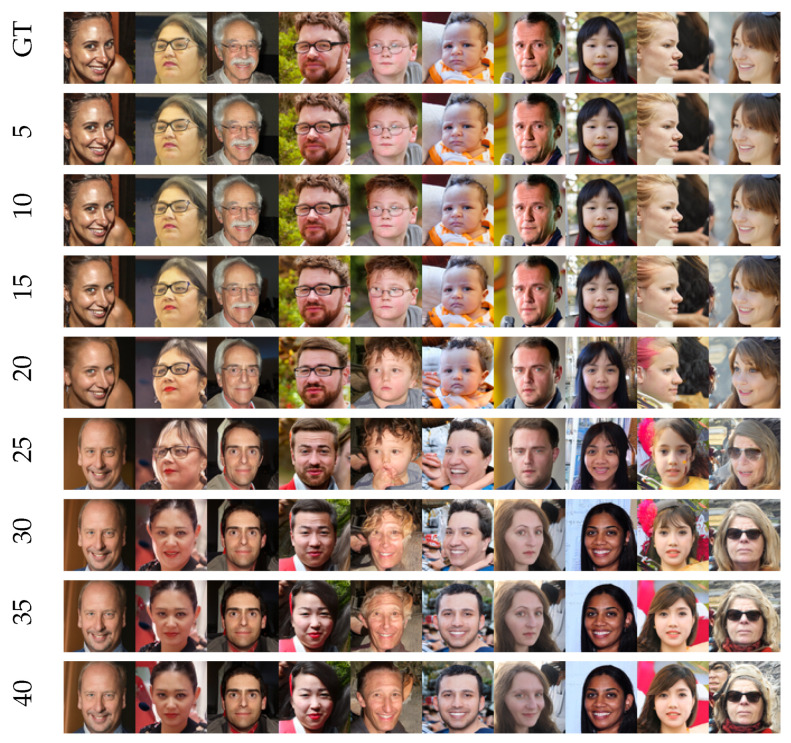
Visual illustration of the U-Turn at different $T_{u}$ in the scheme using the deterministic reverse process (sampler) is presented for the Flickr-Faces-HQ (FFHQ) dataset [5]. This dataset consists of 70,000 high-quality PNG images at 1024 × 1024 resolution and showcases significant variation in terms of age, ethnicity, and image backgrounds. Additionally, the dataset includes a diverse range of accessories such as eyeglasses, sunglasses, hats, and more.

## Data Availability

All data used in this paper are freely available on internet with proper citation in the paper.

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
