# Peer review of "U-Turn Diffusion"

_entropy, 2025, doi:10.3390/e27040343_

Round 1
Reviewer 1 Report
Comments and Suggestions for Authors
This paper introduces a modification to conventional score-based diffusion models that improves generation efficiency while maintaining quality. Through empirical analysis on multiple datasets, the authors identify critical time points in the diffusion process and leverage these insights to shorten both forward and reverse processes. The paper makes contributions through its novel analytical framework, identification of phase transitions in diffusion processes, and the practical U-Turn algorithm that demonstrates promising results. The experimental validation across multiple datasets with both visual and quantitative evaluations strengthens these findings. However, there are the following concerns for publication.
While the paper presents compelling empirical observations about Memorization and Speciation transitions, it lacks rigorous mathematical formulations explaining why these transitions occur. The mechanisms driving these phase transitions are described primarily through experimental observations rather than derived from first principles. I recommend that the authors develop a stronger theoretical framework connecting the observed phenomena to fundamental properties of diffusion processes, or at minimum, explicitly acknowledge these theoretical limitations and outline specific directions for future theoretical work.
The paper would benefit from more comprehensive comparisons with strong baselines, particularly "Elucidating the Design Space of Diffusion Models" (Karras et al., 2022) and other efficient sampling techniques (e.g., DDIM, DPM-Solver). Such comparisons under fixed computational budgets would provide clearer evidence of the practical advantages of the U-Turn approach. I recommend adding comparative experiments that demonstrate where U-Turn excels compared to these established methods, with particular attention to both efficiency and generation quality.
Though testing spans multiple datasets, the paper lacks systematic exploration of how U-Turn parameters vary across different model architectures, data distributions, and training configurations. This raises questions about the robustness of the identified transition points (Tm and Ts). I recommend conducting ablation studies examining the sensitivity of results to architectural choices, noise schedules, and other hyperparameters to demonstrate the broader applicability of the approach beyond the specific experimental settings presented.
Reviewer 2 Report
Comments and Suggestions for Authors This is an interesting well-written math-inclined MS. I support its publication, after some---often cosmetic---revisions, please address the points below. Once an acronym is defined, use it: at least Ground Truth (GT) is defined at least twice. The original papers on the KS test, from 1933 by K and from 1946 by S are to be cited explicitly, in addition to some much later studies on it, such as Ref. [14]. Some of the concepts should be described at more depth, incl. the original citations, to become understandable not only for specialists in the field, but also for a broad community of scientists. For instance, "Fréchet Inception Distance", "Score-Based Diffusion", etc. require this improvement. Already in the intro, please, make it clear to the reader what is new in the current submission regarding the U-turn-diffusion model, as compared to a related material uploaded on the arxiv in 2023. The connections of eq. 4 to the stat-mech condition of Detailed Balance should be described in somewhat more detail. I liked a lot the diffusion-connected part of the MS, but---as not an expert in the image-analysis methods and in albumentation---the sections devoted to the algorithms and the results of the image analysis are much further from my areas of expertise. So, no essential comments will be given to these sections. Still, the connections of the two parts should be presented in more details, in order to give the reader a feeling of a smooth transition from one part of the MS to another. In some images i have seen springboks that reminded me another paper, where the motion of these animals over long times was examined with the help of various diffusion models and also with the machine-learning methods. This Ref. [DOI: https://doi.org/10.1103/PhysRevResearch.5.043129] is rather relevant and can thus be mentioned in the revised version. The authors should describe the actual SPECIFIC outcomes of their proposed algorithm in more detail. The text deteriorates from a bombastic intro to a rather lengthy prose in the main part to finally an abrupt and inconclusive ending. This can certainly be improved so that the whole MS leaves a bombastic impression! The authors should describe in the discussion what are the actual benefits of using this new u-turn-diffusion-based methods for the SPECIFIC tasks of ??? feature-based albumentation, as compared to [i presume] many other methods available on the market, see e.g. Ref. [https://www.mdpi.com/2078-2489/11/2/125]. The section with conclusions is rather weak and must be particularly extended.Author Response
Please see the attache file.

Round 2
Reviewer 1 Report
Comments and Suggestions for Authors
The authors carefully revised the ms, according to the reviewer's comments. The ms now meets the criteria of publication.
Reviewer 2 Report
Comments and Suggestions for Authors
The material can still be improved, the revision is not really satisfactory: one footnote and two paragraphs were added only, as i can see from the blue annotations. If more changes were made, please mark them in blue too upon next revision.